# OculusGraphy: Signal Analysis of the Electroretinogram in a Rabbit Model of Endophthalmitis Using Discrete and Continuous Wavelet Transforms

**DOI:** 10.3390/bioengineering10060708

**Published:** 2023-06-11

**Authors:** Aleksei Zhdanov, Paul Constable, Sultan Mohammad Manjur, Anton Dolganov, Hugo F. Posada-Quintero, Aleksander Lizunov

**Affiliations:** 1Machine Learning and Data Analytics Lab, University of Erlangen-Nuremberg, 91052 Erlangen, Germany; 2Engineering School of Information Technologies, Telecommunications and Control Systems, Ural Federal University Named after the First President of Russia B. N. Yeltsin, 620002 Yekaterinburg, Russia; anton.dolganov@urfu.ru; 3College of Nursing and Health Sciences, Caring Futures Institute, Flinders University, Adelaide, SA 5042, Australia; paul.constable@flinders.edu.au; 4Department of Biomedical Engineering, University of Connecticut, Storrs, CT 06269, USA; sultan_mohammad.manjur@uconn.edu (S.M.M.); h.posada@uconn.edu (H.F.P.-Q.); 5Department of Functional Diagnostics, IRTC Eye Microsurgery Ekaterinburg Center, 620149 Yekaterinburg, Russia; dnmt.oncology@gmail.com

**Keywords:** biomedical research, electroretinography, electroretinogram, ERG, electrophysiology

## Abstract

Background: The electroretinogram is a clinical test used to assess the function of the photoreceptors and retinal circuits of various cells in the eye, with the recorded waveform being the result of the summated response of neural generators across the retina. Methods: The present investigation involved an analysis of the electroretinogram waveform in both the time and time–frequency domains through the utilization of the discrete wavelet transform and continuous wavelet transform techniques. The primary aim of this study was to monitor and evaluate the effects of treatment in a New Zealand rabbit model of endophthalmitis via electroretinogram waveform analysis and to compare these with normal human electroretinograms. Results: The wavelet scalograms were analyzed using various mother wavelets, including the Daubechies, Ricker, Wavelet Biorthogonal 3.1 (bior3.1), Morlet, Haar, and Gaussian wavelets. Distinctive variances were identified in the wavelet scalograms between rabbit and human electroretinograms. The wavelet scalograms in the rabbit model of endophthalmitis showed recovery with treatment in parallel with the time-domain features. Conclusions: The study compared adult, child, and rabbit electroretinogram responses using DWT and CWT, finding that adult signals had higher power than child signals, and that rabbit signals showed differences in the a-wave and b-wave depending on the type of response tested, while the Haar wavelet was found to be superior in visualizing frequency components in electrophysiological signals for following the treatment of endophthalmitis and may give additional outcome measures for the management of retinal disease.

## 1. Introduction

### 1.1. Electroretinogram

The electroretinogram (ERG) is a functional test of the retina that is used clinically [1] to principally assess the function of the photoreceptors and the retinal circuits of horizontal, bipolar, Müller, amacrine, and ganglion cells. The relative contribution of each cell depends upon the state of retinal adaptation, flash color, strength, duration, and stimulus frequency [2]. In standard full-field ERG, the recorded waveform is the result of the summated response of the neural generators across the entire retina. Granit first elucidated that the ERG waveform was the sum of parts through analysis of the dark-adapted (DA) ERG in a series of animal studies using increasing degrees of anesthesia [3]. The ERG waveform typically consists of an initial negative trough termed the ‘a-wave’, which is followed by a positive peak known as the ‘b-wave’. The a-wave represents hyperpolarization of the photoreceptors (rods or cones depending on the degree of retinal adaptation) [4,5,6]. The b-wave has many neural contributors, including the depolarization of the bipolar cells [7,8], Muller cell glial potassium currents [9], and inhibitory horizontal cells [10,11], that all contribute to the main b-wave peak amplitude and timing. In addition, under light-adapted (LA) conditions, the a-wave is also shaped by contributions from post-receptoral neurons [12,13,14]. Inhibitory horizontal cells provide feedback inhibition to photoreceptors under LA and DA conditions [10,15]. The descending limb of the b-wave is mainly shaped by ganglion cells and forms a photopic negative response [16] that is not considered in this analysis. Finally, on the ascending limb of the b-wave, two to three small peaks are observed, which are known as the oscillatory potentials (OPs) that originate in amacrine cells with contributions from bipolar and ganglion cells that contribute to the high-frequency components of the ERG waveform and are typically analyzed after band-pass filtering of the raw ERG signal [10,17,18,19,20]. The rabbit ERG has been used extensively in studies of drug toxicity and diseases of the retina and has a similar shape to the human ERG [7,21,22,23]. This study compares human and rabbit ERGs using signal analysis and the impact of therapeutic treatment of experimental endophthalmitis in rabbits.

### 1.2. Analysis of the Electroretinogram Waveform

To develop and expand clinical applications of the ERG, several strategies have been employed to analyze features of the waveform and its response to different flash strengths. For example, the a-wave’s kinetics have been mathematically modeled [24,25,26] and the OPs have been analyzed using the integrated root-mean-square of their amplitudes [1] and wavelet analyses [27,28]. The modeling of the ERG b-wave amplitude changes with luminance, which is termed the ’photopic hill’, under LA conditions [29,30] and as a more complex luminance response function under DA [31] conditions; this has also been used in clinical studies [32,33,34]. However, these DA and LA luminance response series require several flash strengths and are time-consuming and provide—in the case of the photopic hill—an estimate of the relative contributions of the ON and OFF pathways [30] that may be more readily evaluated using signal analysis of the ERG [35].

Wavelet analysis has been widely used to study ERG in the field of ophthalmology. In recent publications (shown in Table 1), the selection of the mother wavelet has been motivated by various factors, with optimization techniques employed to minimize scatter in the results, leading to improved accuracy in analyzing normal adults’ ERG waveforms [36]. Different wavelets emphasize distinct signal features, making the choice of the most appropriate mother wavelet crucial. Previous studies have identified the Ricker wavelet as the best fit for adult ERG analysis due to its conformity to the waveform shape [37,38]. These studies have successfully addressed classification problems and provided frequency pattern estimates for ERG.

Use of the Morlet wavelet transform in ERG analysis has demonstrated its ability to provide comprehensive insights into the data. For instance, it has been employed for quantifying the frequency, peak time, and power spectrum of the OP components of adult ERGs, offering more informative results compared to other wavelet transforms [43]. Furthermore, Morlet wavelet transforms, along with, potentially, continuous wavelet transform (CWT), have been used to classify glaucomatous and healthy sectors based on frequency content differences in adult ERGs, aiding in the accurate diagnosis and treatment of optic nerve diseases [42]. In the context of pediatric and adult ERG semi-automatic parameter extraction, the Gaussian wavelet has been preferred due to its convenience and better time-domain properties [47]. However, challenges remain in achieving simultaneous localization in both the frequency and time domains, necessitating further advancements in wavelet analysis techniques.

In the paper [46], the analysis of pattern electroretinogram (PERG) signals in the domain of discrete wavelet transform (DWT) coefficients was investigated. The authors chose the DWT as the transform method due to its capability to capture both time and frequency information simultaneously. The mother wavelet utilized in this study was the Daubechies wavelet. The selection of the Daubechies wavelet was based on its desirable properties, such as compact support and orthogonality, which enabled the extraction of relevant features from PERG signals associated with early primary open-angle glaucoma.

The paper [48] explores the application of time–frequency analysis techniques using the DWT and matching pursuits for glaucoma detection. The DWT was selected as the transform method due to its effective capture of both time and frequency information. The mother wavelet employed in this study was the Fejer–Korovkin wavelet, which was chosen for its frequency localization and ability to extract relevant features from ERGs for glaucoma diagnosis.

The study [45] investigated the use of wavelet decomposition analysis in two-flash multifocal ERG for early glaucoma detection. The authors employed wavelet decomposition, a chosen DWT transform method based on the Daubechies wavelet, to capture both time and frequency information in the ERG signals. The study compares the wavelet decomposition analysis to ganglion cell analysis and visual field tests, highlighting its potential as a valuable diagnostic tool.

The main application of signal analysis to the ERG waveform was developed by Gauvin, who applied a discrete wavelet transform (DWT) analysis to extract energies corresponding to the ON and OFF pathways of the a- and b-waves as well as the early and later OPs [35,42,49]. The DWT application of Gauvin’s work has been applied to the identification of neurodevelopmental and retinal disease [45,48,50,51] and offers the potential to develop the scope of practice for ERGs [52]. This study contributes to this field by utilizing discrete and continuous wavelet analyses of human and rabbit ERGs to provide additional insights into the patterns of ERG changes during recovery from endophthalmitis in rabbits that complement traditional time-domain features usually reported for ERGs, such as the amplitudes and time-to-peaks of the main a- and b-waves.

Here, we analyzed the ERG waveform in the time and time–frequency domains using DWT and CWT to monitor the effects of treatment in an experimental rabbit model of endophthalmitis. This application of signal analysis may provide additional and complementary information about the underlying neural generators of the ERG and supports the advancement of the use of the ERG as a clinical test of retinal and central nervous system function [52,53]. The application of signal analysis has been widely used to evaluate biological signals in cardiac [54] and EEG recordings [55,56]; however, this approach is relatively uncommon in ophthalmology, with DWT used to evaluate glaucoma [45,46,48] and neurodevelopmental disorders [42,48]. Here, we compare CWT with DWT in the analysis of recovery from an experimental model of endophthalmitis that has not been used previously.

## 2. Materials and Methods

### 2.1. Study Protocol and Signals

All recordings were made with the Tomey GmBH EP-1000 stimulator sampling at 2 kHz with a 0.1–300 Hz bandpass filter. Two flash strengths were used: DA 2.0 (scotopic response) and the LA 2.0 (maximal response). Flash stimuli were a white 2 cd·s·m${}^{-2}$ on a 0 cd·s·m${}^{-2}$ background. Loop electrodes positioned in the lower lid were used for human recordings, and ERG–Jet lens electrodes were used for the rabbit recordings; they have an excellent an signal-to-noise ratio [57]. Oxybuprocaine 0.4% was used as an anesthetic, with Mydraicyl used to dilate the pupils in human and rabbit subjects. The use of Mydraicyl for pupil dilation has been well-documented in human and animal studies, and its safety and efficacy have been established. In addition, the short duration of action of Mydraicyl makes it a preferred choice for ophthalmic procedures where rapid onset and quick recovery of normal function are desired. The experimental protocols are described in Table 2, with all signals amplified and digitized using the EP-1000 system’s built-in amplifier and A/D converter [58].

Figure 1 shows the normal ERG signals from healthy human and rabbit subjects. The scotopic 2.0 ERG response is shown in Figure 1a, and the maximum 2.0 ERG response is shown in Figure 1b. Table 3 provides information on the age of subjects, as well as the values of the following parameters: *a*—amplitude of the a-wave, *b*—amplitude of the b-wave, *la*—latency of the a-wave, and *lb*—latency of the b-wave.

Given the small sample size, a qualitative comparison of the a- and b-wave time-domain parameters are presented. The individual values of the maximum LA 2.0 ERG responses are presented in Table 3, which shows that the amplitudes of the a-waves of adults and children were, respectively, 5.4 and 4.3 times higher than those in rabbits. The b-wave amplitudes were, respectively, 3.6 and 1.9 times larger in humans compared to rabbits. The time-to-peaks of the a- and b-waves differed by no more than 5.4% between human and rabbit recordings. The individual values of the scotopic DA 2.0 ERG response parameters are shown in Table 3, with the amplitudes of the b-wave of human adults and children 67.34% and 52.82% larger, respectively, than the recorded b-wave amplitudes of the rabbits. Analysis of the a-wave was not performed due to the weak response of the photoreceptors at this flash strength. The average values of the time-to-peak of the b-wave of adults and children relative to the time-to-peak of the b-wave of the rabbits were 52.88% and 34.68% slower, respectively.

To further explore the time–frequency domain analysis of the ERG waveforms, we used DWT and CWT analysis using the following mother wavelets:1.Daubechies wavelet;2.Ricker wavelet;3.Wavelet Biorthogonal 3.1 (bior3.1);4.Morlet wavelet;5.Haar wavelet;6.Gaussian wavelet.

These mother wavelets have previously been used for the analysis of ERG waveforms, and thus, we selected this series of wavelets with which to perform this analysis [59]. We included these six wavelets to determine the most-suitable mother wavelet for the maximum 2.0 ERG responses and scotopic 2.0 ERG responses used in this study.

Figure 2 illustrates the processing pipelines for CWT and DWT. The CWT processing pipeline involved a series of steps depicted in the flow diagram.

To analyze the original ERG waveforms, we employed CWT with various mother wavelets. The frequency range investigated in this study spanned from 0.5 Hz to 200 Hz with an even logarithmic spacing. The outcome of the CWT process yielded the wavelet scalogram, which was subsequently analyzed using the previously described segment analysis technique.

The DWT processing pipeline, as shown in the flow diagram, encompassed the following steps. Initially, the original ERG waveforms were upsampled from 1000 Hz to 1280 Hz. Next, DWT was performed on the upsampled signal, employing frequency bands centered on 10, 20, 40, 80, and 160 Hz. The resulting scalogram was then analyzed using fixed time and frequency segments.

### 2.2. Endophthalmitis Treatment in Rabbit

Twelve male New Zealand rabbits weighing between 3.0 to 3.2 kg (mean ± SD) 3.10 ± 0.35 kg and aged 2.0 to 3.5 months were treated for bacterial endophthalmitis of the right eye associated with staphylococcus aureus infection. The right eye of each rabbit was injected intravitreally with a solution of quantum dots 0.01% dose of QD InP/ZnSe/ZnS 660 in combination with vancomycin 10 mg/mL as a 0.1 mL solution [60]. Article [60] provides an exploration of the evaluation of ophthalmotoxicity stemming from quantum-dot-based bioconjugates pertaining to their potential in the treatment of resistant endophthalmitis.

During the study, slit-lamp photography (SLP) of the anterior segment, optical coherence tomography (OCT) of the posterior segment, and ERG recordings were recorded. SLP was performed using a Haag–Streit BQ 900 slit lamp manufactured by Haag–Streit Holding AG. OCT was performed using an Optovue RTVue-100 manufactured by Optovue Inc. The research involving laboratory animals was conducted in strict adherence to the principles outlined in the code of ethics, specifically the “Directive 2010/63/EU of the European Parliament and the Council of the European Union”.

OCT, SLP, and ERG measurements and recordings were performed weekly. The maximum 2.0 ERG response and scotopic 2.0 ERG response recordings are shown in Figure 3a,b. Figure 3, respectively. During the treatment period, improvements were observed in the amplitudes of the a- and b-waves. Table 4 provides the values of the following parameters: *a*—amplitude of the a-wave, *b*—amplitude of the b-wave, *la*—latency of the a-wave, and *lb*—latency of the b-wave.

The scotopic 2.0 ERG response results presented in Table 4 during three weeks of treatment show gradual recovery of the main time-domain parameters of the ERG waveforms. The qualitative values *a*, *b*, *la*, and *lb* recorded at week 3 show an improvement from the average values of healthy rabbits by 26.09%, 12.28%, 11.67%, and 4.88%, respectively. The maximum 2.0 ERG response results are presented in Table 4 and also demonstrate the therapeutic effect of the treatment. The value of the *b* and *lb* recorded at week 3 also show a qualitative improvement compared to the control of 32.00% and 20.46%, respectively.

## 3. Results

### 3.1. Wavelet Scalograms of Human and Rabbit ERGs

Figure 4 and Figure 5 show the wavelet scalograms of the maximum 2.0 ERG response and the scotopic 2.0 ERG response. The CWT scalograms depict the time–frequency representation, where the horizontal axis denotes time in milliseconds (ms) and the vertical axis represents the frequency values in Hertz (Hz). The Gaussian eight-degree wavelet figure also displays the segment numbering for the corresponding scalogram. To describe the CWT wavelet scalograms uniformly, the scalogram areas were divided into even and odd segments as previously described [47]. The spatial arrangement and energy of central segments Nos. 1–4 were evaluated because these segments contain the main physiological information and have a low level of noise associated with them that relate to blinks and eye-movement artifacts. The figures presented in this study utilize black and white arrows to delineate regions of interest with heightened information content. Additionally, the wavelet scalogram segments are labeled numerically as 1–6, providing a systematic reference for analysis. Notably, the intensity of the color (red) within the scalogram corresponds to the magnitude of signal strength, with brighter shades indicating greater amplitude in the respective area.

The figures presented in this study utilize black and white arrows to delineate regions of interest with heightened information content. Additionally, the wavelet scalogram segments are labeled numerically as 1–6, providing a systematic reference for analysis. Notably, the intensity of the color (red) within the scalogram corresponds to the magnitude of signal strength, with brighter shades indicating greater amplitude in the respective area.

The DWT wavelet scalograms represent the energy (µV·s) within the signal that was extracted for statistical analysis. The DWT scalogram represents the energies within each frequency band centered on 20, 40, 80, and 160 Hz. The horizontal axis of the DWT scalogram denotes time (in milliseconds), and the vertical axis represents frequency bands in Hertz (Hz). It should be noted that the signal was upsampled from 1000 to 1280 Hz to obtain the 20–160 Hz frequency bands.

The CWT (Ricker wavelet) wavelet scalogram in Figure 4 demonstrates significant energy and localization differences between the rabbit and human signals. The area with maximum energy in Adult 1 is in the range of 0–45 ms and 3–67 Hz (segment No. 1), 50–100 ms and 3–67 Hz (segment No. 2); Child 1—18–45 ms and 5–65 Hz (segment No. 1), 56–87 ms and 5–65 Hz (segment No. 2); and Rabbit 1—55–65 ms and 8–20 Hz (segment No. 2).

The maximum energies for the CWT (Morlet wavelet) wavelet scalograms are as follows: Adult 1—0–45 ms and 5–50 Hz (segment No. 1), 50–100 ms and 5–50 Hz (segment No. 2); Child 1—18–45 ms and 7–50 Hz (segment No. 1), 56–80 ms and 7–50 Hz (segment No. 2); and Rabbit 1—60–70 ms and 7–20 Hz (segment No. 2).

The maximum energies for the CWT (Gaussian eight-degree wavelet) wavelet scalograms are as follows: Adult 1—0–50 ms and 5–50 Hz (segment No. 1), 50–100 ms and 5–50 Hz (segment No. 2); Child 1—18–45 ms and 5–50 Hz (segment No. 1), 55–90 ms and 5–50 Hz (segment No. 2); and Rabbit 1—12–37 ms and 7.8–15 Hz (segment No. 1), 55–68 ms and 5–23 Hz (segment No. 2).

The DWT (Daubechies wavelet) wavelet scalogram in Figure 4 demonstrates most of the energy in the lower frequency regions: Adult 1—0–100 samples for 10 Hz (maximum energy), 50–100 samples for 20 Hz (medium energy), from 40 Hz (low energy); Child 1—0–100 samples for 10 Hz (maximum energy), 50–100 samples for 20 Hz and 100–200 samples for 10 Hz (medium energy), from 40 Hz (low energy); and Rabbit 1—100–200 samples for 10 Hz (maximum energy), from 40 Hz (low energy).

The results for the DWT (Biorthogonal 3.1 wavelet) wavelet scalograms are: Adult 1—0–100 samples for 10 Hz (maximum energy), 0–50 samples for 20 Hz (medium energy), from 40 Hz (low energy); Child 1–0–100 samples for 10 Hz (maximum energy), 0–50 samples for 20 Hz (medium energy), from 40 Hz (low energy); and Rabbit 1—0–100 samples for 10 Hz (maximum energy), from 20 Hz (low energy).

The results for the DWT (Haar wavelet) wavelet scalograms are: Adult 1—0–100 samples for 10 Hz (maximum energy), 0–50 samples for 20 Hz and 50–100 samples for 40 Hz (medium energy), from 40 Hz (low energy); Child 1—0–100 samples for 10 Hz and 0–50 samples for 20 Hz (maximum energy), 50–100 samples for 20 Hz and about 70 samples for 40 Hz (medium energy), from 80 Hz (low energy); and Rabbit 1—0–100 samples for 10 Hz (maximum energy), 100–200 samples for 10 Hz (medium energy), from 20 Hz (low energy).

The CWT (Ricker wavelet) wavelet scalogram in Figure 5 shows noticeable variations in energy levels and locations between the signals produced by rabbits and humans. The area with maximum energy in Adult 1 is in the range of 0–130 ms and 1–37 Hz (segment No. 1), 150–250 ms and 1–37 Hz (segment No. 2); Child 1—0–130 ms and 1–37 Hz (segment No. 1), 50–250 ms and 1–10 Hz (segment No. 2); and Rabbit 1—20–70 ms and 100–200 Hz (segment No. 1), 2–20 ms and 3–16 Hz (segment No. 2).

The results for the CWT (Morlet wavelet) wavelet scalograms are: Adult 1—0–130 ms and 2–16 Hz (segment No. 1), 130–250 ms and 2–16 Hz (segment No. 2); Child 1—0–140 ms and 2–16 Hz (segment No. 1), 140–250 ms and 2–14 Hz (segment No. 2); and Rabbit 1—25–75 ms and 3–16 Hz (segment No. 1), 100–150 ms and 3–16 Hz (segment No. 2).

The results for the CWT (Gaussian eight-degree wavelet) wavelet scalograms are: Adult 1—0–50 ms and 5–50 Hz (segment No. 1), 50–100 ms and 5–50 Hz (segment No. 2); Child 1—18–45 ms and 5–50 Hz (segment No. 1), 55–90 ms and 5–50 Hz (segment No. 2); and Rabbit 1—12–37 ms and 7.8–15 Hz (segment No. 1), 55–68 ms and 5–23 Hz (segment No. 2).

The DWT (Daubechies wavelet) wavelet scalogram in Figure 5 shows the majority of the energy is concentrated in the lower frequency regions: Adult 1—100–300 samples for 10 Hz (maximum energy), 450–600 samples for 10 Hz (medium energy), from 20 Hz (low energy); Child 1—100–200 samples for 10 Hz (maximum energy), 200–600 samples for 10 Hz (medium energy), from 20 Hz (low energy); and Rabbit 1—0–50 samples for 10 Hz (maximum energy), 200–300 samples for 10 Hz (medium energy), from 20 Hz (low energy).

The results for the DWT (Biorthogonal 3.1 wavelet) wavelet scalograms are: Adult 1—0–200 samples for 10 Hz (maximum energy), 300–600 samples for 10 Hz (medium energy), from 20 Hz (low energy); Child 1—0–50 samples for 10 Hz (maximum energy), 100–600 samples for 10 Hz (medium energy), from 20 Hz (low energy); and Rabbit 1—100–200 samples for 10 Hz (maximum energy), 200–300 samples for 10 Hz (medium energy), from 20 Hz (low energy).

The results for the DWT (Haar wavelet) wavelet scalograms are: Adult 1—0–100 samples for 10 Hz (maximum energy), 400–600 samples for 100 Hz (medium energy), from 20 Hz (low energy); Child 1—0–100 samples for 10 Hz and 300–600 samples for 10 Hz (medium energy), from 20 Hz (low energy); and Rabbit 1—0–150 samples for 10 Hz and 100–150 samples for 20 Hz (maximum energy), 150–400 samples for 10 Hz (medium energy), from 30 Hz (low energy).

### 3.2. Endophthalmitis Treatment in Rabbit

The CWT (Ricker wavelet) wavelet scalogram in Figure 6 shows treatment responses over 3 weeks. The area with the maximum energy in the 1st week is in the range of 0–130 ms and 2–37 Hz (segment No. 1), 170–250 ms and 1–5 Hz (segment No. 2); 2nd week—55–70 ms and 5–50 Hz (segment No. 1); and 3rd week—0–120 ms and 2–17 Hz (segment No. 1), 130–250 ms and 2–15 Hz (segment No. 2).

The CWT (Morlet wavelet) wavelet results are: 1st week—80–120 ms and 7–18 Hz (segment No. 1), 130–170 ms and 7–18 Hz (segment No. 2); 2nd week—25–80 ms and 2–55 Hz (segment No. 1), 80–175 ms and 2–55 Hz (segment No. 2); and 3rd week—0–80 ms and 2–18 Hz (segment No. 1), 110–200 ms and 2–18 Hz (segment No. 2).

The CWT (Gaussian eight-degree wavelet) wavelet results are: 1st week—55–70 ms and 5–11 Hz (segment No. 1), 140–160 ms and 6–11 Hz (segment No. 2); 2nd week—60–65 ms and 9–14 Hz (segment No. 1), 97–102 ms and 9–14 Hz (segment No. 2); and 3rd week—0–87 ms and 2–18 Hz (segment No. 1), 112–200 ms and 2–18 Hz (segment No. 2).

The DWT (Daubechies wavelet) wavelet scalogram in Figure 6 indicates that most of the energy was focused in the lower frequency range: 1st week—100–200 samples, 400 amples for 10 Hz (maximum energy), 50–200 samples for 20 Hz (medium energy), from 40 Hz (low energy); 2nd week—0–100 samples for 10 Hz (maximum energy), from 20 Hz (low energy); and 3rd week—0–100 samples for 10 Hz (maximum energy), 100–400 samples for 10 Hz (medium energy), from 20 Hz (low energy).

The DWT (Biorthogonal 3.1 wavelet) wavelet results are: 1st week—0–2100 samples for 10 Hz (maximum energy), 250–350 samples for 10 Hz (medium energy), from 20 Hz (low energy); 2nd week—100–200 samples, 400 samples for 10 Hz and 0–50 samples for 20 Hz (maximum energy), 250–350 samples for 10 Hz (medium energy), from 40 Hz (low energy); and 3rd week—0–100 samples for 10 Hz (maximum energy), 100–400 samples for 10 Hz (medium energy), from 20 Hz (low energy).

The DWT (Haar wavelet) wavelet results are: 1st week—0–100 samples, 250–350 samples for 10 Hz (maximum energy), from 20 Hz (low energy); 2nd week—100–350 samples for 10 Hz (medium energy), from 80 Hz (low energy); and 3rd week—0–100 samples for 10 Hz (maximum energy), 250–400 samples for 10 Hz (medium energy), from 20 Hz (low energy).

It should be noted that the ERG in the 1st week showed a signal shift to the right. The ERG region with maximum energy in the 3rd week corresponded to the frequency characteristics of the rabbit ERG in Figure 4; however, they have time extensions of 50% (segment No. 1) and 30% (segment No. 2). In addition, the high-frequency components of the signal are more pronounced compared to Figure 4.

CWT (Ricker wavelet) wavelet scalogram in Figure 7 shows treatment for 3 weeks. The area with maximum energy in the 1st week is in the range of 25–130 ms and 2–37 Hz (segment No. 1), 150–250 ms and 2–16 Hz (segment No. 2); 2nd week—25–130 ms and 2–37 Hz (segment No. 1); and 3rd week—0–130 ms and 2–30 Hz (segment No. 1), 130–250 ms and 2–16 Hz (segment No. 2).

The CWT (Morlet wavelet) wavelet results are: 1st week—75–130 ms and 5–16 Hz (segment No. 1), 150–175 ms and 5–16 Hz (segment No. 2); 2nd week—scalogram segments are not differentiable and represent sets of spots or discontinuities in the signal corresponding to different scales and frequencies; and 3rd week—0–130 ms and 2–16 Hz (segment No. 1), 130–250 ms and 2–16 Hz (segment No. 2).

The CWT (Gaussian eight-degree wavelet) wavelet results are: 1st week—4–10 ms and 90–110 Hz (segment No. 1); 2nd week—scalogram segments are not differentiable and represent sets of spots or discontinuities in the signal corresponding to different scales and frequencies; and 3rd week—1–17 ms and 0–110 Hz (segment No. 1), 1–17 ms and 0–110 Hz (segment No. 2)

The DWT (Daubechies wavelet) wavelet scalogram in Figure 7 indicates that the majority of the energy is concentrated in the lower frequency range: 1st week—0–50 samples for 10 Hz (maximum energy), 50–100 samples for 20 Hz (medium energy), from 40 Hz (low energy); 2nd week—0–50 samples for 10 Hz and 100–150 samples for 20 Hz (maximum energy), 100–200 samples for 10 Hz (medium energy), from 20 Hz (low energy); and 3rd week—0–50 samples for 10 Hz (maximum energy), 50–200 samples for 10 Hz (medium energy), from 20 Hz (low energy).

The DWT (Biorthogonal 3.1 wavelet) wavelet results are: 1st week—550–600 samples for 10 Hz (maximum energy), 200–300 samples for 10 Hz (medium energy), from 20 Hz (low energy); 2nd week—0–50 samples for 10 Hz (maximum energy), 100–150 samples for 20 Hz (medium energy), from 40 Hz (low energy); and 3rd week—0–50 samples for 10 Hz (maximum energy), 200–600 samples for 10 Hz (medium energy), from 20 Hz (low energy).

The DWT (Haar wavelet) wavelet results are: 1st week—550–600 samples for 10 Hz (maximum energy), 200–550 samples for 10 Hz (medium energy), from 20 Hz (low energy); 2nd week—0–50 samples for 10 Hz (maximum energy), 100–200 samples for 20 Hz (medium energy), from 40 Hz (low energy); and 3rd week—0–50 samples for 10 Hz and 150–200 samples for 20 Hz (maximum energy), 200–600 samples for 10 Hz (medium energy), from 40 Hz (low energy).

The results obtained from CWT indicate that the pathological states exhibit a marked differentiation of segments, leading to the formation of low-energy patches during inflammation. On the other hand, DWT analysis demonstrated a non-linear and irregular distribution of signal frequency components during inflammation. One limitation of this preliminary report is that we were unable to conduct a full quantitative correlation with OCT parameters such as retinal nerve fiber layer thickness owing to an inability to perform accurate segmentation of the retinal scans. Future studies in human subjects with active retinitis may allow for a fuller description of ocular structural and functional changes observed during resolution of ocular inflammation examined using ERG signal analysis. In addition, such ERG analyses may also support further studies where rabbits are used as a model for endophthalmitis.

Figure 8 shows the OCT and SLP images of rabbits at weekly intervals during recovery from endophthalmitis secondary to bacterial infection. There are improvements in the structures of the anterior segment of the eye during treatment according to SLP. Preservation of the normal anatomy of the sensory retina of the posterior segment of the eye is observed. Hyper-reflective dots in the preretinal region were only visible in the 1st week of active endophthalmitis as imaged using OCT.

The results from the 1st week show: Large regions of hyperreflective elements and preretinal macular membrane formation were visualized in the posterior segment of the eye using OCT scans. In turn, pericorneal injection and increased iris pastosity along with newly formed vessels of the iris were visualized in the anterior segment of the eye using SLP.

The results from the 2nd week show: Decreased hyperreflective elements allowed for assessing the photoreceptors’ destruction articulation and the pigment epithelium–Bruch’s membrane complex was now visible with decreased iris pastosity along with a number of new vessels in the anterior segment of the eye.

The results from the 3rd week show: Normalization of the posterior segment of the eye was visualized except for a single foci of photoreceptor damage. The condition of the anterior segment of the eye corresponded to the control eye with resolution of new blood vessels and edema.

Thus, the results SLP and OCT confirm that the rabbit ERG showed functional improvement that was consistent with the resolution of endophthalmitis as observed using OCT and SLP.

## 4. Discussion

The mother wavelets used in the study are similar in form and parameters to the ERG; in particular, their impulsive natures are similar. For example, the difference between the Morlet wavelet and the Gaussian wavelet is that the Morlet wavelet has better selectivity properties in the frequency domain, while the Gaussian wavelet has better properties in the time domain. However, it is proving elusive to obtain localization of an ideal character simultaneously in the frequency and time domains.

The basis functions of CWT, in most cases, are not strictly orthogonally normalized due to the fact that the elements of the basis are infinitely differentiated with exponential characteristics falling off at time infinity. DWT does not have the above problems due to the higher accuracy of the reconstruction of the studied signals. The choice of the type and typology of wavelets primarily depends on the studied ERG signal, and the task of this analysis takes into account the level of knowledge and experience of the researcher. However, DWT may be advantageous when analyzing signals within the a- and b-wave discrete time windows and provides a direct measure of the underlying ON and OFF pathways in the retina [35].

Therefore, it is necessary to pay attention to verification and determination of the level of the biomedical signal’s effectiveness and whether or not physiological interpretation or a more general picture of the energy contained across the whole waveform depicted when using CWT is required. It should also be noted that when using similar ERG analysis of other protocols, different results may be obtained in practical terms.

In the given context, the comparison of adult, child, and rabbit ERG responses using CWT showed that the energy of the adult signals in segments No. 1 and 2 were significantly higher than those of the child signals, which is consistent with the larger responses recorded in the adult subjects. Additionally, the rabbit signals exhibited a less-pronounced a-wave in segment No. 1 and a less-pronounced b-wave in segment No. 2, with the magnitude of the effect being dependent on the state of retinal adaption and amplitude of the response. The differences in the spectral compositions of the ERGs between rabbit and human are of note and may be the result of differences in retinal morphology and may have implications for studies involving rabbit models of retinal pathology [61,62]. The Morlet wavelet and Gaussian eight-degree wavelet are commonly used in signal processing applications and have been found to have similar performance in extracting parameters from the scalogram, providing a visual representation of a signal’s time–frequency components. However, in the analysis of electrophysiological signals such as adult, child, and rabbit signals, it has been observed that the DWT tends to exhibit low-frequency components due to the presence of software filtering in the electrophysiology equipment used. In contrast, the Haar wavelet has been found to provide superior visualization of frequency components in such signals.

Experimental studies in ophthalmology are often performed on the eyes of rabbits due to their similar size and the ease of making clinical observations without additional specialized equipment [63,64]. On the other hand, the rabbit retina has significant differences from the human retina (merangiotic type of the retina, absence of the macular area, and different distribution of photoreceptors of various types) [65,66]. Nevertheless, the results obtained in this study demonstrate the possibility of quantifying functional changes in the time-domain and time–frequency-domain representations in rabbit ERGs under scotopic and photopic conditions. However, caution should be taken, as the features observed using signal analysis show differences from human ERGs despite a similar shape under photopic and scotopic conditions. These signal analytical methods may provide further insights into the recovery stages of retinal inflammation in conjunction with clinical features to improve patient management. Further studies in human subjects with retinitis would be required to further quantify ERG signal analysis markers with structural changes using OCT.

Previous studies using ERGs during endophthalmitis in humans have relied upon time-domain measures such as the b/a wave amplitude [67] or a- and b-wave amplitudes in rabbit models of bacterial endophthalmitis [68] that may not always show differences despite active inflammation [69,70]. Hence, there is a need for additional markers that may be derived from the ERG. In addition, given the development of novel therapeutic drug delivery systems such as nanoparticles for the treatment of endophthalmitis, the analysis of features extracted from the ERG may be beneficial as biomarkers in clinical trials [71].

## 5. Conclusions

The findings of this work offer valuable insights into the effects of treatment on the scotopic 2.0 ERG and maximum 2.0 ERG responses in rabbits, showcasing a gradual recovery of the main time-domain parameters of the ERG waveforms during a three-week treatment period. The qualitative values, including *a*, *b*, *la*, and *lb*, exhibited significant improvements compared to the average values of healthy rabbits, providing evidence of the therapeutic effect of the treatment. The positive outcomes were further supported by the maximum 2.0 ERG response results, which showed enhancements in the *b* and *lb* parameters.

Analyzing the wavelet scalograms of human and rabbit ERGs provided additional understanding of the energy and localization patterns of the signals, revealing distinct characteristics in the frequency and time domains for each species. These differences can be attributed to physiological variations and signal processing mechanisms between rabbits and humans. The wavelet scalogram analysis enabled a comprehensive assessment of the treatment response and identified specific areas of energy concentration at different time intervals.

Furthermore, the treatment for endophthalmitis in rabbits demonstrated promising outcomes in terms of the wavelet scalograms, with shifts in energy distribution and frequency characteristics observed over the three-week treatment period. The energy levels and locations in the scalograms indicated the treatment’s effectiveness in restoring the physiological properties of the ERG signals, including time extension and enhanced high-frequency components compared to the baseline ERG characteristics.

Overall, this work highlights the potential of wavelet analysis in evaluating the treatment response of ERG signals in rabbits, providing valuable information on the therapeutic effects of the treatment on the scotopic 2.0 ERG and maximum 2.0 ERG responses. The observed improvements in the time-domain parameters and wavelet scalograms suggest a restoration of retinal function and support the efficacy of the treatment in treating endophthalmitis. These findings contribute to the advancement of ophthalmic research and may have implications for future treatment approaches in both animal models and human patients.

## Figures and Tables

**Figure 1 bioengineering-10-00708-f001:**
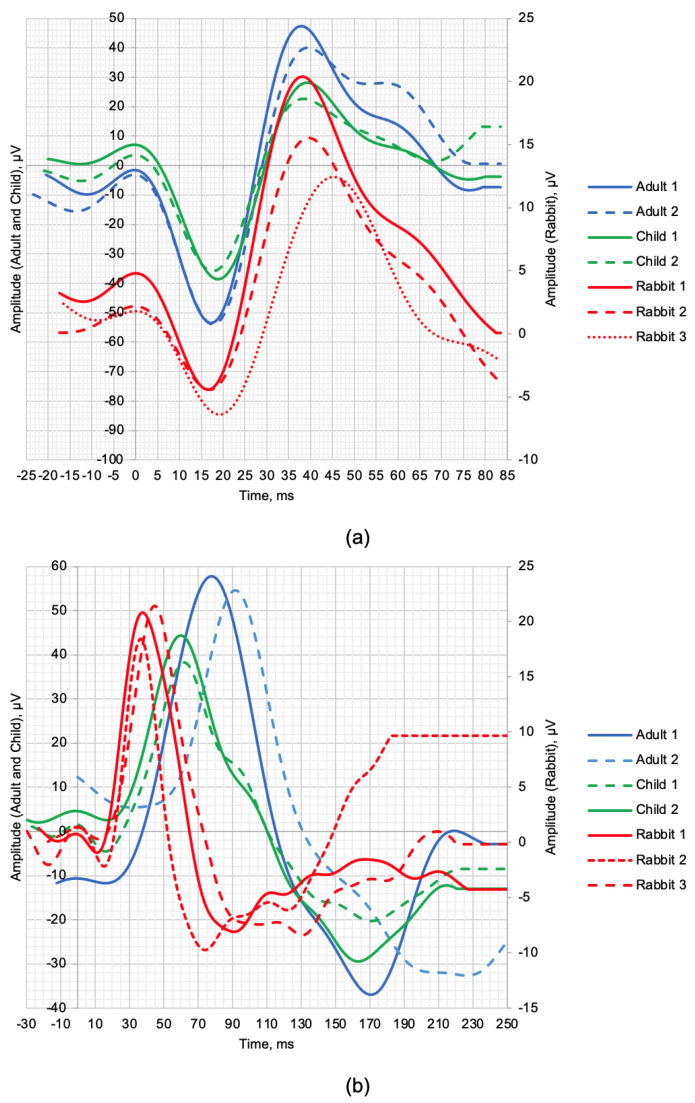
ERG signal time−domain representation of healthy human subjects and rabbits: (**a**) scotopic 2.0 ERG response and (**b**) maximum 2.0 ERG response.

**Figure 2 bioengineering-10-00708-f002:**
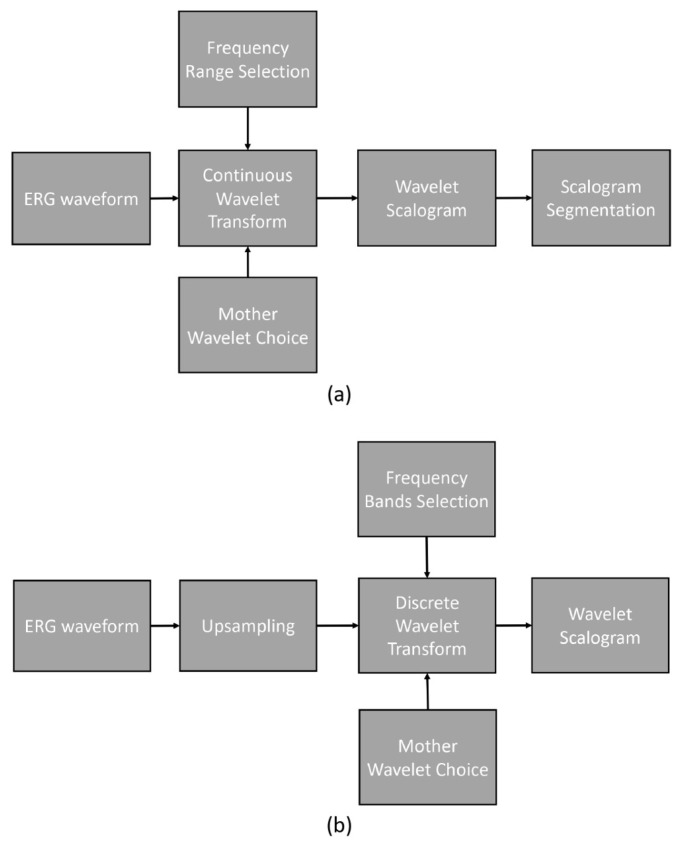
Processing pipelines: (**a**) continuous wavelet transform and (**b**) discrete wavelet transform.

**Figure 3 bioengineering-10-00708-f003:**
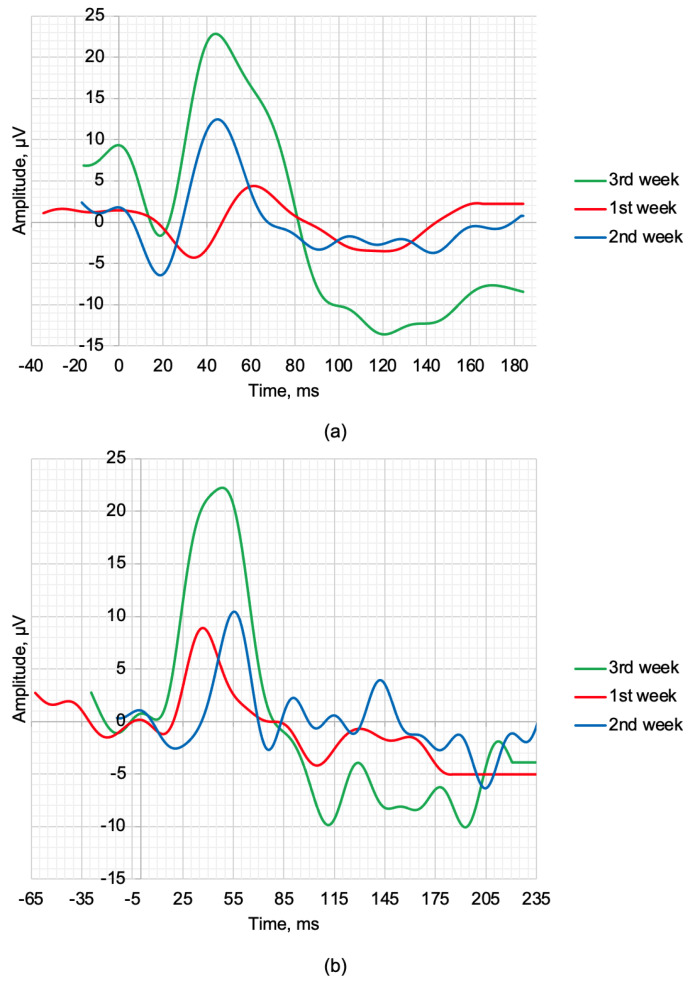
ERG waveforms in the time−domain representation of rabbits recorded at weekly intervals following treatment: (**a**) scotopic 2.0 ERG response and (**b**) maximum 2.0 ERG response.

**Figure 4 bioengineering-10-00708-f004:**
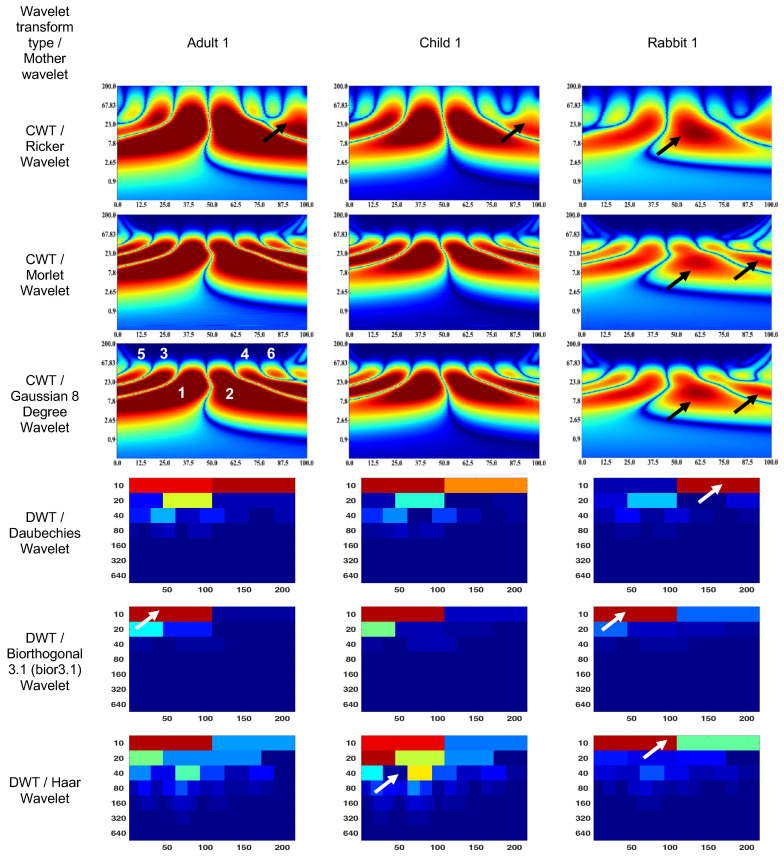
Wavelet scalograms of maximum 2.0 ERG response in healthy human and rabbit subjects.

**Figure 5 bioengineering-10-00708-f005:**
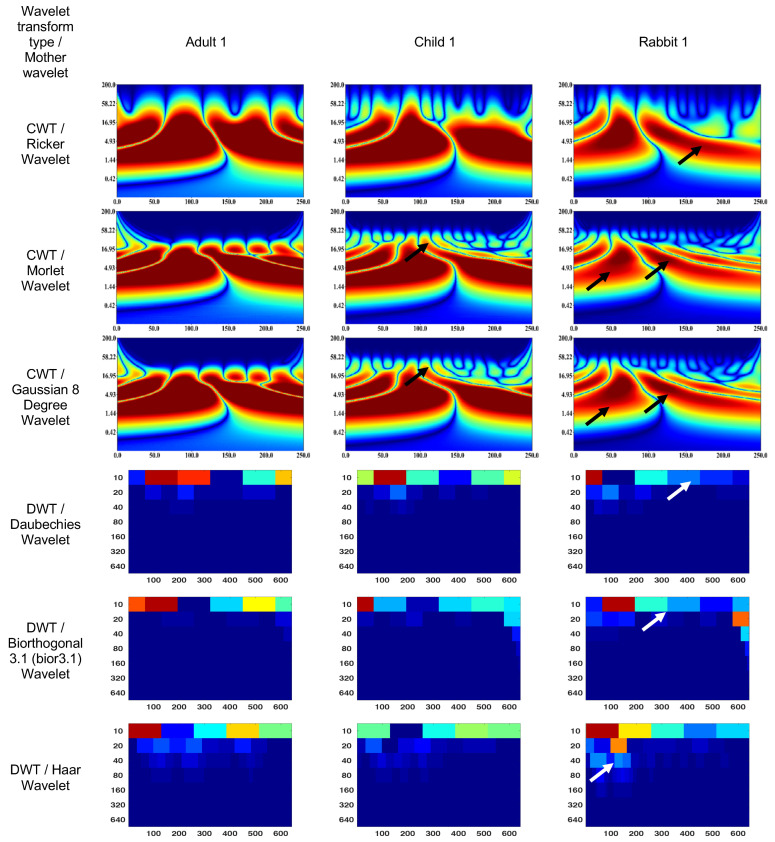
Wavelet scalograms of scotopic 2.0 ERG response waveforms in healthy human and rabbit subjects.

**Figure 6 bioengineering-10-00708-f006:**
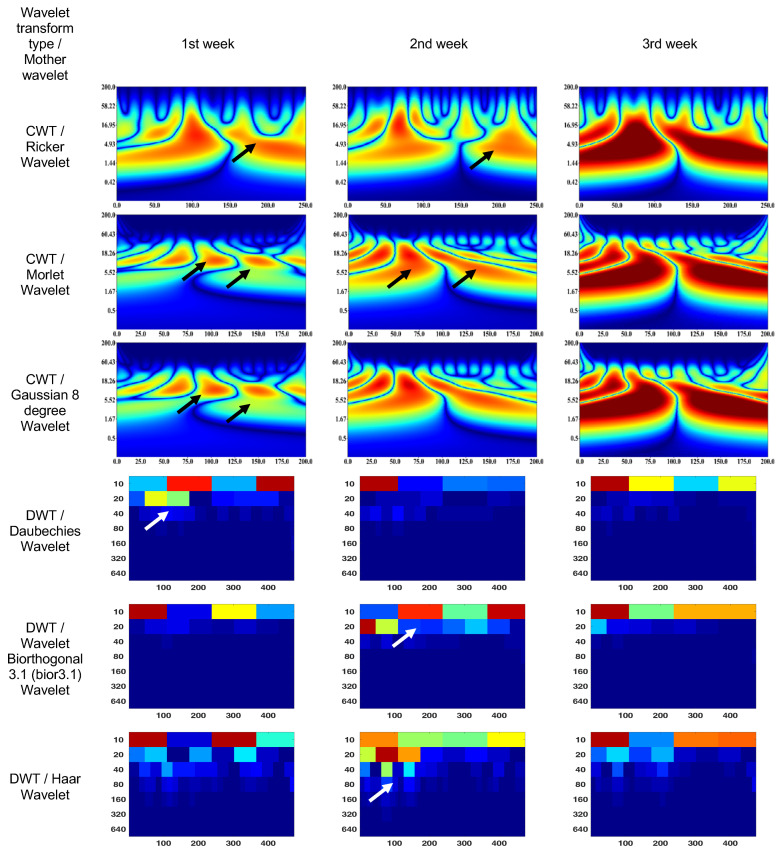
Wavelet scalograms of maximum 2.0 ERG response in rabbit following treatment at weeks 1, 2, and 3.

**Figure 7 bioengineering-10-00708-f007:**
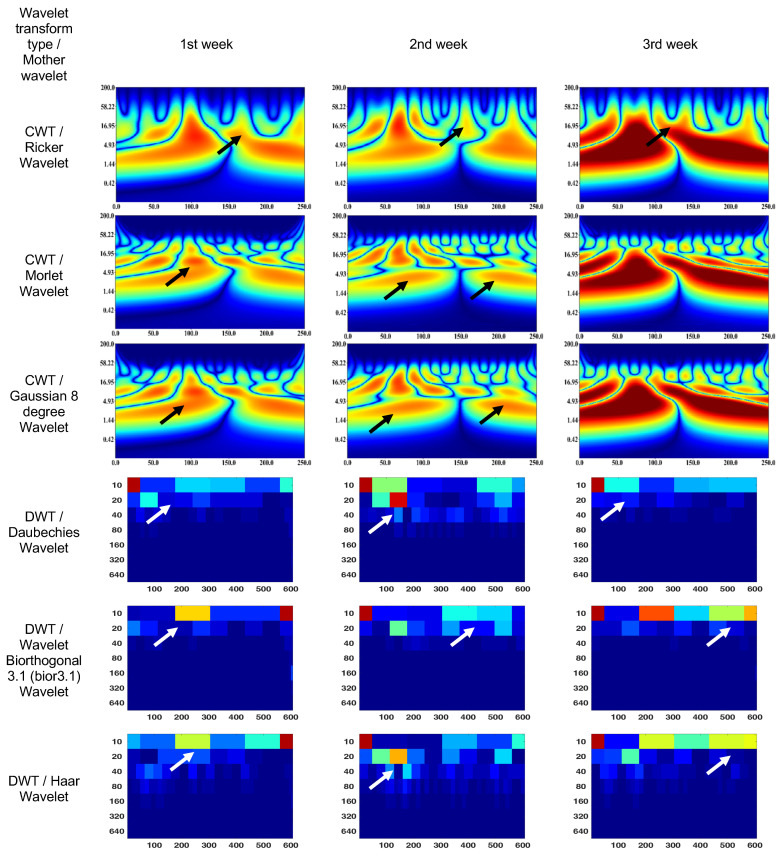
Wavelet scalograms of scotopic 2.0 ERG response in rabbit following treatment at weeks 1, 2 and 3.

**Figure 8 bioengineering-10-00708-f008:**
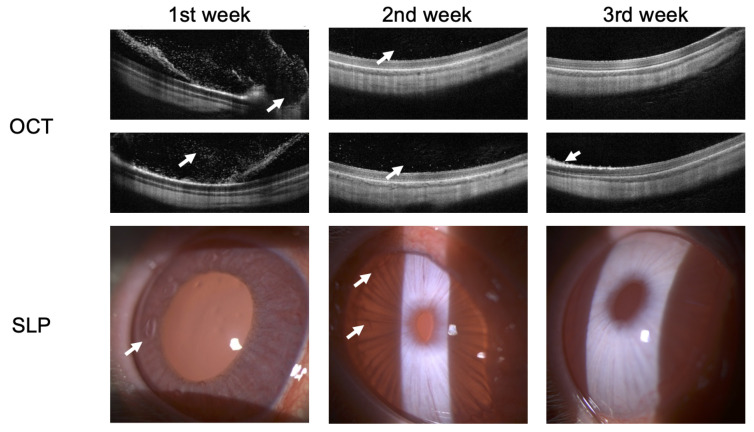
OCT and SLP images of rabbit at weekly intervals during recovery of endophthalmitis. Arrows indicate the main anatomical changes following treatment.

**Table 1 bioengineering-10-00708-t001:** Recent studies on wavelet analysis in ERG.

Year	Author	Transform Type (Mother Wavelet)	Signals (Subjects)
2005	Penkala [37]	CWT (Morlet, Ricker)	120 (N/A)
2007	Penkala [36]		102 (N/A)
2010	Barraco [39]	CWT (Ricker)	24 (N/A)
2011	Barraco [40]		N/A (10)
2011	Barraco [41]		N/A (10)
2014	Gauvin [42]	CWT (Morse), DWT (Haar)	N/A (40)
2014	Dimopoulos [43]	CWT (Morlet)	N/A (63)
2015	Miguel-Jiménez [44]		N/A (47)
2017	Brandao [45]	DWT (Daubechies)	60 (60)
2019	Hassankarimi [46]	DWT (Daubechies)	60 (N/A)
2020	Ahmadieh [38]	CWT (Morlet)	N/A (36)
2022	Zhdanov [47]	CWT (Gaussian)	425 (N/A)
2022	Sarossy [48]	DWT (Fejer–Korovkin)	103 (55)

**Table 2 bioengineering-10-00708-t002:** Characteristics of an electrophysiological study.

	Scotopic 2.0 ERG Response	Maximum 2.0 ERG Response
Flash intensity, cd·s·m${}^{-2}$	2	2
Background light, cd·s·m${}^{-2}$	0	0
Flash duration, ms	0.5	3
Flash frequency, kHz	20	20
Stimulus interval, s (cps)	2.5 (0.4)	10 or 13 (0.1)
Flash color	white	white

**Table 3 bioengineering-10-00708-t003:** Scotopic 2.0 ERG response and maximum 2.0 ERG response parameters of healthy human and rabbit subjects.

	Adult 1	Adult 2	Child 1	Child 2	Rabbit 1	Rabbit 2	Rabbit 3
Age	20.5 y.o.	27.8 y.o.	10.9 y.o.	7.5 y.o.		2–3.5 mos.	
Maximum 2.0 ERG Response							
*a*, µV	52.32	50.69	45.65	39.28	9.18	6.59	8.2
*b*, µV	101.22	93.78	66.74	58.35	24.75	19.92	18.86
*la*, ms	17.5	18	19	18	16.5	17	19.5
*lb*, ms	38	39.5	39.5	38.5	38	39.5	45.5
Scotopic 2.0 ERG Response							
*a*, µV	1.04	-	6.05	2.03	1.82	3.47	1.04
*b*, µV	69.53	60.01	42.77	46.91	21.87	20.47	21.12
*la*, ms	17	-	16	16.5	10.5	14.5	12.5
*lb*, ms	78	92.5	62.5	60.5	38.5	37.5	44.5

**Table 4 bioengineering-10-00708-t004:** Scotopic 2.0 ERG response and maximum 2.0 ERG response parameters of rabbits recorded at weekly intervals following treatment.

	1st Week	2nd Week	3rd Week
Maximum 2.0 ERG Response			
*a*, µV	5.61	8.03	10.81
*b*, µV	16.48	18.6	24.14
*la*, ms	33	17.5	20
*lb*, ms	47	46.5	43
Scotopic 2.0 ERG Response			
*a*, µV	1.81	3.43	0.75
*b*, µV	10.26	12.72	21.92
*la*, ms	13	22.5	8.5
*lb*, ms	39.5	57.5	50.5

## Data Availability

Zhdanov, A.E.; Dolganov, A.Y.; Borisov, V.I.; Lucian, E.; Bao, X.; Kazaijkin, V.N.; Ponomarev, V.O.; Lizunov, A.V.; Ivliev, S.A. OculusGraphy: Pediatric and Adults Electroretinograms Database, 2020. https://doi.org/10.21227/y0fh-5v04 (accessed on 29 November 2022).

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
