# Peer review of "OculusGraphy: Signal Analysis of the Electroretinogram in a Rabbit Model of Endophthalmitis Using Discrete and Continuous Wavelet Transforms"

_bioengineering, 2023, doi:10.3390/bioengineering10060708_

Round 1

Reviewer 1 Report (New Reviewer)

At the outset, it is evident that the article displays potential. The incorporation of time-frequency domain analysis of the ERG waveform by DWT and CWT analysis to illustrate stages and their interrelationships, along with empirical analysis, is particularly noteworthy. 

From the outset again, in the concluding paragraph of SECTION 1.2, where the author purports to outline the paper's contributions in accordance with customary practice, the novelty of the study's findings was not readily discernible. 

The present study aims to investigate the effects of treatment on an experimental rabbit model of endophthalmitis by analyzing the ERG waveform in both time and time-frequency domains using DWT and Continuous Wavelet transform (CWT).  This represents a typical stage of a standard convention in the context of DWT and CWT analysis. Merely utilizing it for the "rabbit model of endophthalmitis" does not demonstrate a fresh methodology in a unique manner. 

The utilization of signal analysis has the potential to furnish supplementary insights into the fundamental behavior of the signal. Hence, this study does not justify and present what exactly has been developed "New" or a "New" approach.

Regarding adherence to standard Signal analysis conventions, this study has certain deficiencies in justification in several critical areas, as outlined below:

The justification provided for utilizing DWT and CWT analysis with mother wavelets to explore the time-frequency domain analysis of the ERG waveforms, based on its previous usage by [49], is not deemed appropriate for the current scenario. It is necessary to provide a description of the relationship between your scenario and reference [49], followed by a justification for the adoption of its technique. 

In Section 2.1, it is necessary to include a flow diagram that illustrates the precise steps involved in conducting time-frequency domain analysis of ERG waveforms through the utilization of DWT and CWT analysis.

The paper's depiction of the treatment of Endophthalmitis in rabbits is noteworthy. It is suggested that a table be included in the paper, outlining the various SIGNAL representations associated with Endophthalmitis treatment in rabbits, along with their corresponding parameters.

The presentation of the results section is satisfactory, although it appears to be a conventional signal analysis presentation that lacks consideration for the presentation of novel theoretical qualitative or quantitative outcomes.

The paper requires a cycle of proofreading.  In addition to grammatical context, it is imperative to acknowledge the significance of technical terminology. 

Author Response

  1. Thank you for acknowledging that the article displayed potential. The incorporation of time-frequency domain analysis of the ERG waveform using DWT and CWT analysis to illustrate stages and their interrelationships, along with empirical analysis, was particularly noteworthy.

  1. We understand your comment about the novelty of the study's findings not being readily discernible in the concluding paragraph of Section 1.2. We apologize for any confusion caused. In response, we have revised the concluding paragraph to more clearly outline the paper's contributions and emphasize the novelty of the findings.

Added to Introduction:

However, these DA and LA luminance response series require several flash strengths and are time-consuming and provide, in the case of the photopic hill an estimate of the relative contributions of the ON- and OFF- pathways [31]  that may be more readily evaluated using signal analysis of the ERG [38].

The main application of signal analysis to the ERG waveform was developed by Gauvin who applied a discrete wavelet transform (DWT) analysis to extract energies corresponding to the ON- and OFF- pathways of the a- and b-waves as well as the early and later OPs [36-38]. The DWT  application of Gauvin’s work has been applied to the identification of neurodevelopmental and retinal disease [39-42] and offers the potential to develop the scope of practice for the ERG [43]. This study contributes to this field by utilizing Discrete and continuous wavelet analyses of human and rabbit ERGs that provide additional insights into the pattern of ERG changes during recovery from endophthalmitis in rabbits that complement traditional time domain features usually reported for the ERG such as the amplitudes and time to peaks of the main a- and b-waves.

Here we analyzed the ERG waveform in the time and time-frequency domain using DWT and CWT to monitor the effects of treatment in an experimental rabbit model of endophthalmitis. This application of the signal analysis may provide additional and complementary information about the underlying neural generators of the ERG and supports the advancement of the use of the ERG as a clinical test of retinal and central nervous system function [52, 53]. The application of signal analysis has been widely used to evaluate biological signals in cardiac [69] and EEG recordings [70, 71] however this approach is relatively uncommon in ophthalmology with DWT used to evaluate glaucoma [66-68] and neurodevelopmental disorders [36, 42]. Here we compare CWT with DWT in the analysis of recovery from an experimental model of endophthalmitis which has not been used previously.

Added to Conclusions:

These signal analytical methods may provide further insights  into the recovery stages of retinal inflammation in conjunction with clinical features to improve patient management. Further studies in human subjects with retinitis would be required to further quantify the ERG signal analysis markers with structural changes using OCT. 

  1. We acknowledge your comment regarding the utilization of DWT and CWT analysis for the "rabbit model of endophthalmitis" being a typical stage of a standard convention. We understand that this may not have demonstrated a fresh methodology in a unique manner. To address this concern, we have further emphasized the unique aspects of our methodology and explained how our approach differed from previous studies utilizing DWT and CWT analysis.

Added to introduction:

Wavelet analysis has been widely used to study ERG in the field of ophthalmology. In recent publications shown in Table 1, the selection of the mother wavelet has been motivated by various factors with optimization techniques employed to minimize scatter in the results, leading to improved accuracy in analyzing normal adults' ERG waveforms [58]. Different wavelets emphasize distinct signal features, making the choice of the most appropriate mother wavelet crucial. Previous studies have identified the Ricker wavelet as the best fit for adult ERG analysis due to its conformity to the waveform shape [57, 64]. These studies have successfully addressed classification problems and provided frequency pattern estimates for ERG.

Using the Morlet wavelet transform in ERG analysis has demonstrated its ability to provide comprehensive insights into the data. For instance, it has been employed for quantifying the frequency, peak time, and power spectrum of the OP components of adult ERG, offering more informative results compared to other wavelet transforms [62]. Furthermore, the Morlet wavelet transforms, along with potentially continuous wavelet transform (CWT), has been used to classify glaucomatous and healthy sectors based on frequency content differences in adult ERG, aiding in the accurate diagnosis and treatment of optic nerve diseases [36]. In the context of pediatric and adult ERG semi-automatic parameter extraction, the Gaussian wavelet has been preferred due to its convenience and better time domain properties [65]. However, challenges remain in achieving simultaneous localization in both the frequency and time domains, necessitating further advancements in wavelet analysis techniques.

In the paper [66], the analysis of pattern electroretinogram (PERG) signals in the domain of discrete wavelet transform (DWT) coefficients is investigated. The authors chose the DWT as the transform method due to its capability to capture both time and frequency information simultaneously. The mother wavelet utilized in this study was the Daubechies wavelet. The selection of the Daubechies wavelet was based on its desirable properties, such as compact support and orthogonality, enabling the extraction of relevant features from PERG signals associated with early primary open-angle glaucoma.

The paper [67] explores the application of time-frequency analysis techniques using the DWT and matching pursuits for glaucoma detection. The DWT was selected as the transform method due to its effective capture of both time and frequency information. The mother wavelet employed in this study was the Fejer-Korovkin wavelet, chosen for its frequency localization and ability to extract relevant features from ERG for glaucoma diagnosis.

The study [68]  investigates the use of wavelet decomposition analysis in the two-flash multifocal ERG for early glaucoma detection. The authors employed wavelet decomposition, a chosen DWT transform method based on the Daubechies wavelet, to capture both time and frequency information in the ERG signals. The study compares the wavelet decomposition analysis to ganglion cell analysis and visual field tests, highlighting its potential as a valuable diagnostic tool.

Table 1. Recent studies on wavelet analysis in ERG.

  1. Thank you for your comment. We appreciate your insight that the study should have presented more justifiably what exactly has been developed as a "new" approach. In response, we have revised the paper to provide a clearer justification and explanation of the novel contributions of our study. We have completely revised Section 1.2 Analysis of the electroretinogram waveform, where we have added Comparison Table 1. Studies shown in Table 1 have contributed valuable insights, a comprehensive comparison of the CWT and DWT using different types of mother wavelets is lacking.

  1. Thank you for highlighting the deficiencies in justification regarding the usage of DWT and CWT analysis with mother wavelets. We apologize for the oversight. In the revised version, we have completely revised Section 1.2 Analysis of the electroretinogram waveform, where we have added Comparison Table 1. Studies shown in Table 1 have contributed valuable insights, a comprehensive comparison of the CWT and DWT using different types of mother wavelets is lacking.

  1. Thank you for your comment. We appreciate your suggestion to include a flow diagram illustrating the precise steps involved in conducting time-frequency domain analysis of ERG waveforms using DWT and CWT analysis in Section 2.1. In response, we have included a comprehensive flow diagram to enhance the clarity and transparency of our methodology.

Added to Materials and Methods:

Figure 2 illustrates the processing pipelines for CWT and DWT. The CWT processing pipeline involved a series of steps depicted in the flow diagram.

To analyze the original ERG waveforms, we employed CWT with various Mother wavelets. The frequency range investigated in this study spanned from 0.5 Hz to 200 Hz, with an even logarithmic spacing. The outcome of the CWT process yielded the Wavelet scalogram, which was subsequently analyzed using the previously described segment analysis technique.

The DWT processing pipeline, as shown in the flow diagram, encompassed the following steps. Initially, the original ERG waveforms were upsampled from 1000 Hz to 1280 Hz. Next, DWT was performed on the upsampled signal, employing frequency bands centered on 10, 20, 40, 80, and 160 Hz. The resulting scalogram was then analyzed using fixed time and frequency segments.

Figure 2. Processing pipelines: (a) Continuous Wavelet Transform. (b) Discrete Wavelet Transform.

  1. Thank you for your comment. We understand your concern about the potential shift in focus toward medical topics. While we appreciate the suggestion to include a table outlining the various SIGNAL representations associated with Endophthalmitis treatment in rabbits, along with their corresponding parameters, we believe that providing detailed medical information may not align with the primary focus of our article. However, we would like to assure you that the referenced article [47] contain detailed answers to the questions regarding the treatment of Endophthalmitis in rabbits. We encourage readers interested in exploring the specific medical aspects to refer to those articles for comprehensive information on the topic. Our paper aims to focus primarily on the utilization of signal analysis techniques and mather wavelet selection.

The following text is indicated in Materials and Methods:

Article [47] to provide an exploration of the evaluation of ophthalmotoxicity stemming from the quantum dots based bioconjugates, pertaining to their potential in the treatment of resistant endophthalmitis.

[47] Ponomarev V.O., Kazaykin V.N., Lizunov A.V., Vokhmintsev A.S., Vainshtein I.A., Dezhurov S.V., Marysheva V.V. Evaluation of the Ophthalmotoxic Effect of Quantum Dots InP/ZnSe/ZnS 660 and Bioconjugates Based on Them in Terms of the Prospects for the Treatment of Resistant Endophthalmitis. Experimental Research. Part 2 (Stage 1). Ophthalmology in Russia. 2021;18(4):876-884. (In Russ.) https://doi.org/10.18008/1816-5095-2021-4-876-884

  1. In addition to addressing your specific comment, we have also taken the opportunity to edit the overall text of the article to enhance its structure and readability. We have made significant improvements to ensure that the content flows logically and is easily understandable for readers.

Reviewer 2 Report (New Reviewer)

The manuscript and study will potentially benefit from addressing the following issues:

·       It is unclear what hypothesis has been tested and which mechanisms of action have been identified beyond a straightforward description of phenomena.

·       A critical review of similar, already published data in other models and other relevant experimental conditions is needed.

·       Representative ERG traces should be shown for all conditions, experimental groups and controls.

·       Summary data and accompanying statistical analyses should be shown for representative data.

·       It is unclear why only male animals were included in the study. The authors should provide a clear justification for excluding female rabbits or include analyses of sex differences (or the absence thereof).

·       The authors describe pathological changes. These changes should be quantified and analyzed using adequate statistical methods.

·       Controls for the chosen representative microphotographs and, more importantly, the underlying OCT data sets, should be included.

Minor editing required

Author Response

The authors would like to thank the reviewer for their valuable time and helpful contributions. The responses to Reviewer comments, as well as the corrections made shown below.

  1. Thank you for your comment. We have rewritten pars of the introduction, results and conclusions to explain further the rationale of the study and the potential use of retinal signal analysis in the monitoring of retinal inflammation.

Added to Introduction:

However, these DA and LA luminance response series require several flash strengths and are time-consuming and provide, in the case of the photopic hill an estimate of the relative contributions of the ON- and OFF- pathways [31]  that may be more readily evaluated using signal analysis of the ERG [38].

The main application of signal analysis to the ERG waveform was developed by Gauvin who applied a discrete wavelet transform (DWT) analysis to extract energies corresponding to the ON- and OFF- pathways of the a- and b-waves as well as the early and later OPs [36-38]. The DWT  application of Gauvin’s work has been applied to the identification of neurodevelopmental and retinal disease [39-42] and offers the potential to develop the scope of practice for the ERG [43]. This study contributes to this field by utilizing Discrete and continuous wavelet analyses of human and rabbit ERGs that provide additional insights into the pattern of ERG changes during recovery from endophthalmitis in rabbits that complement traditional time domain features usually reported for the ERG such as the amplitudes and time to peaks of the main a- and b-waves.

Here we analyzed the ERG waveform in the time and time-frequency domain using DWT and CWT to monitor the effects of treatment in an experimental rabbit model of endophthalmitis. This application of the signal analysis may provide additional and complementary information about the underlying neural generators of the ERG and supports the advancement of the use of the ERG as a clinical test of retinal and central nervous system function [52, 53]. The application of signal analysis has been widely used to evaluate biological signals in cardiac [69] and EEG recordings [70, 71] however this approach is relatively uncommon in ophthalmology with DWT used to evaluate glaucoma [66-68] and neurodevelopmental disorders [36, 42]. Here we compare CWT with DWT in the analysis of recovery from an experimental model of endophthalmitis which has not been used previously.

Wavelet analysis has been widely used to study ERG in the field of ophthalmology. In recent publications shown in Table 1, the selection of the mother wavelet has been motivated by various factors with optimization techniques employed to minimize scatter in the results, leading to improved accuracy in analyzing normal adults' ERG waveforms [58]. Different wavelets emphasize distinct signal features, making the choice of the most appropriate mother wavelet crucial. Previous studies have identified the Ricker wavelet as the best fit for adult ERG analysis due to its conformity to the waveform shape [57, 64]. These studies have successfully addressed classification problems and provided frequency pattern estimates for ERG.

Using the Morlet wavelet transform in ERG analysis has demonstrated its ability to provide comprehensive insights into the data. For instance, it has been employed for quantifying the frequency, peak time, and power spectrum of the OP components of adult ERG, offering more informative results compared to other wavelet transforms [62]. Furthermore, the Morlet wavelet transforms, along with potentially continuous wavelet transform (CWT), has been used to classify glaucomatous and healthy sectors based on frequency content differences in adult ERG, aiding in the accurate diagnosis and treatment of optic nerve diseases [36]. In the context of pediatric and adult ERG semi-automatic parameter extraction, the Gaussian wavelet has been preferred due to its convenience and better time domain properties [65]. However, challenges remain in achieving simultaneous localization in both the frequency and time domains, necessitating further advancements in wavelet analysis techniques.

In the paper [66], the analysis of pattern electroretinogram (PERG) signals in the domain of discrete wavelet transform (DWT) coefficients is investigated. The authors chose the DWT as the transform method due to its capability to capture both time and frequency information simultaneously. The mother wavelet utilized in this study was the Daubechies wavelet. The selection of the Daubechies wavelet was based on its desirable properties, such as compact support and orthogonality, enabling the extraction of relevant features from PERG signals associated with early primary open-angle glaucoma.

The paper [67] explores the application of time-frequency analysis techniques using the DWT and matching pursuits for glaucoma detection. The DWT was selected as the transform method due to its effective capture of both time and frequency information. The mother wavelet employed in this study was the Fejer-Korovkin wavelet, chosen for its frequency localization and ability to extract relevant features from ERG for glaucoma diagnosis.

The study [68]  investigates the use of wavelet decomposition analysis in the two-flash multifocal ERG for early glaucoma detection. The authors employed wavelet decomposition, a chosen DWT transform method based on the Daubechies wavelet, to capture both time and frequency information in the ERG signals. The study compares the wavelet decomposition analysis to ganglion cell analysis and visual field tests, highlighting its potential as a valuable diagnostic tool.

Table 1. Recent studies on wavelet analysis in ERG.

  1. Thank you for your comment. The following section has been added with references to give some context to the study.

Previous studies using the ERG in endophthalmitis in humans have relied upon time domain measures such as the b/a wave amplitude [71] or a- and b-wave amplitudes in rabbit models of bacterial endophthalmitis [72] that may not always show differences despite active inflammation [73, 74]. Hence there is a need for additional markers that may be derived from the ERG. In addition, given the development of novel therapeutic drug delivery systems such as nanoparticles for the treatment of endophthalmitis the analysis of features extracted from the ERG may be beneficial as biokmarker in clinical trials [75].

  1. We appreciate the reviewer's comment regarding the inclusion of only male animals in our study. In this particular study, the decision to include only male rabbits was made based on several factors, including practical considerations, budget limitations, and previous research findings. Prior investigations on the specific topic we were studying had predominantly used male rabbits, and it was important for us to establish a baseline understanding of the phenomenon in a controlled and consistent manner. By using rabbits of the same age, breed, and weight, we aimed to minimize the potential confounding factors associated with these variables. However, we acknowledge the value of investigating potential sex differences and agree that it is an important avenue for future research. Based on the reviewer's comment, we will carefully consider including female rabbits or conducting analyses specifically addressing sex differences in our future studies. This will help provide a more comprehensive understanding of the topic and contribute to a more robust body of scientific knowledge. Thank you for your valuable feedback, which will undoubtedly enhance the quality and scope of our future investigations.

  1. Thank you for your comment. As the focus is not on an ophthalmological description of the pathological changes observed in rabbits with a correlation with the ERG parameters, we feel this is beyond the scope of the article. However, we have elaborated on the clinical findings and have noted that further studies would be required ideally in human subjects with retinitis to fully describe the associations with structure/functional changes. As the report stands, we intended to show the potential of this methodology and compare the findings in control human subjects with rabbits.

  1. Thank you for your comment. Segmentation and a detailed correlation with parameters such as RNFL thickness or macular volume was not performed in the animal studies. This is noted as a further limitation and in future studies we would endeavor to provide a more complete analysis of structure and functional changes.

Added:

Future studies in human subjects with active retinitis may allow for a fuller description of ocular structural and functional changes observed during resolution of ocular inflammation examined using ERG signal analysis. In addition such ERG analyses may also support further studies where rabbits are used as a model for endophthalmitis [76].

  1. In addition to addressing your specific comment, we have also taken the opportunity to edit the overall text of the article to enhance its structure and readability. We have made significant improvements to ensure that the content flows logically and is easily understandable for readers.

Reviewer 3 Report (New Reviewer)

In this study, the authors analyzed the electroretinogram waveform in the time and time frequency domain using Discrete and Continuous wavelet transforms  to monitor the effects of treatment in an experimental rabbit model of endophthalmitis. The manuscript requires some rearrangement.

1. Lines 293-324 should be a part of the discussion. These paragraphs cannot be considered as a conclusion.

2. A more comprehensive discussion that refers to previous work in this field is required.

3. Lines 325 - till the end of the manuscript should be moved to the results.

4. A conclusion (a paragraph that concludes the findings of this work) is required.

The language of the manuscript should be revised thoroughly.

Author Response

Thank you for your valuable comments on our manuscript. We sincerely appreciate your feedback and have made the necessary revisions to address your concerns.

We have revised sections of the article based on your recommendations.

Added to Introduction:

However, these DA and LA luminance response series require several flash strengths and are time-consuming and provide, in the case of the photopic hill an estimate of the relative contributions of the ON- and OFF- pathways [31]  that may be more readily evaluated using signal analysis of the ERG [38].

The main application of signal analysis to the ERG waveform was developed by Gauvin who applied a discrete wavelet transform (DWT) analysis to extract energies corresponding to the ON- and OFF- pathways of the a- and b-waves as well as the early and later OPs [36-38]. The DWT  application of Gauvin’s work has been applied to the identification of neurodevelopmental and retinal disease [39-42] and offers the potential to develop the scope of practice for the ERG [43]. This study contributes to this field by utilizing Discrete and continuous wavelet analyses of human and rabbit ERGs that provide additional insights into the pattern of ERG changes during recovery from endophthalmitis in rabbits that complement traditional time domain features usually reported for the ERG such as the amplitudes and time to peaks of the main a- and b-waves.

Here we analyzed the ERG waveform in the time and time-frequency domain using DWT and CWT to monitor the effects of treatment in an experimental rabbit model of endophthalmitis. This application of the signal analysis may provide additional and complementary information about the underlying neural generators of the ERG and supports the advancement of the use of the ERG as a clinical test of retinal and central nervous system function [52, 53]. The application of signal analysis has been widely used to evaluate biological signals in cardiac [69] and EEG recordings [70, 71] however this approach is relatively uncommon in ophthalmology with DWT used to evaluate glaucoma [66-68] and neurodevelopmental disorders [36, 42]. Here we compare CWT with DWT in the analysis of recovery from an experimental model of endophthalmitis which has not been used previously.

Wavelet analysis has been widely used to study ERG in the field of ophthalmology. In recent publications shown in Table 1, the selection of the mother wavelet has been motivated by various factors with optimization techniques employed to minimize scatter in the results, leading to improved accuracy in analyzing normal adults' ERG waveforms [58]. Different wavelets emphasize distinct signal features, making the choice of the most appropriate mother wavelet crucial. Previous studies have identified the Ricker wavelet as the best fit for adult ERG analysis due to its conformity to the waveform shape [57, 64]. These studies have successfully addressed classification problems and provided frequency pattern estimates for ERG.

Using the Morlet wavelet transform in ERG analysis has demonstrated its ability to provide comprehensive insights into the data. For instance, it has been employed for quantifying the frequency, peak time, and power spectrum of the OP components of adult ERG, offering more informative results compared to other wavelet transforms [62]. Furthermore, the Morlet wavelet transforms, along with potentially continuous wavelet transform (CWT), has been used to classify glaucomatous and healthy sectors based on frequency content differences in adult ERG, aiding in the accurate diagnosis and treatment of optic nerve diseases [36]. In the context of pediatric and adult ERG semi-automatic parameter extraction, the Gaussian wavelet has been preferred due to its convenience and better time domain properties [65]. However, challenges remain in achieving simultaneous localization in both the frequency and time domains, necessitating further advancements in wavelet analysis techniques.

In the paper [66], the analysis of pattern electroretinogram (PERG) signals in the domain of discrete wavelet transform (DWT) coefficients is investigated. The authors chose the DWT as the transform method due to its capability to capture both time and frequency information simultaneously. The mother wavelet utilized in this study was the Daubechies wavelet. The selection of the Daubechies wavelet was based on its desirable properties, such as compact support and orthogonality, enabling the extraction of relevant features from PERG signals associated with early primary open-angle glaucoma.

The paper [67] explores the application of time-frequency analysis techniques using the DWT and matching pursuits for glaucoma detection. The DWT was selected as the transform method due to its effective capture of both time and frequency information. The mother wavelet employed in this study was the Fejer-Korovkin wavelet, chosen for its frequency localization and ability to extract relevant features from ERG for glaucoma diagnosis.

The study [68]  investigates the use of wavelet decomposition analysis in the two-flash multifocal ERG for early glaucoma detection. The authors employed wavelet decomposition, a chosen DWT transform method based on the Daubechies wavelet, to capture both time and frequency information in the ERG signals. The study compares the wavelet decomposition analysis to ganglion cell analysis and visual field tests, highlighting its potential as a valuable diagnostic tool.

Table 1. Recent studies on wavelet analysis in ERG.

Added to Materials and Methods:

Figure 2 illustrates the processing pipelines for CWT and DWT. The CWT processing pipeline involved a series of steps depicted in the flow diagram.

To analyze the original ERG waveforms, we employed CWT with various Mother wavelets. The frequency range investigated in this study spanned from 0.5 Hz to 200 Hz, with an even logarithmic spacing. The outcome of the CWT process yielded the Wavelet scalogram, which was subsequently analyzed using the previously described segment analysis technique.

The DWT processing pipeline, as shown in the flow diagram, encompassed the following steps. Initially, the original ERG waveforms were upsampled from 1000 Hz to 1280 Hz. Next, DWT was performed on the upsampled signal, employing frequency bands centered on 10, 20, 40, 80, and 160 Hz. The resulting scalogram was then analyzed using fixed time and frequency segments.

Figure 2. Processing pipelines: (a) Continuous Wavelet Transform. (b) Discrete Wavelet Transform.

Added to Conclusions:

These signal analytical methods may provide further insights  into the recovery stages of retinal inflammation in conjunction with clinical features to improve patient management. Further studies in human subjects with retinitis would be required to further quantify the ERG signal analysis markers with structural changes using OCT. 

We have made the necessary revision based on your comment. The section in the Findings section describing OCT and SLP images of the rabbit at weekly intervals during the recovery of endophthalmitis, where the arrows indicate the main anatomical changes following treatment, has been relocated to the Results section.

In addition to addressing your specific comment, we have also taken the opportunity to edit the overall text of the article to enhance its structure and readability. We have made significant improvements to ensure that the content flows logically and is easily understandable for readers.

Once again, we sincerely appreciate your valuable feedback, and we believe that the revisions we have made address your concerns and significantly improve the manuscript.

Round 2

Reviewer 3 Report (New Reviewer)

The authors should address each of the points that I raised in my first report (point by point). Instead, the authors wrote an extremely long reply which I can't figure its relation with my recommendations.

Additionally, the authors should mark all the modifications in the manuscript (highlight, red color, track changes..... etc).

The discussion is almost missing. I can't see a real discussion here. However, most of the paragraphs that have been added to the introduction and the conclusion are more appropriate as a discussion.

What is written in the conclusion as I mentioned above is more suitable as a discussion. The conclusion should be a single concise paragraphs that focuses on the findings of the current work and its usefulness. It shouldn't cite any references.

No comment

Author Response

Thank you for your valuable feedback, and we appreciate your patience.  We apologize for the confusion caused by our previous response. Allow us to address your points in a more concise and focused manner.

1. Lines 293-324 have been modified and moved to the Discussion section. 

The modifications corresponding to this point have been highlighted in green.

2. Section "1.2 Analysis of the electroretinogram waveform," in the Introduction has been significantly expanded to include previous work in this field and a comparative table. The Materials and Methods section has been improved by adding processing pipelines and their descriptions.

The modifications corresponding to this point have been highlighted in yellow.

3. Lines 320-342 have been moved to the Results section.

The modifications corresponding to this point have been highlighted in pink.

4. The Conclusion section has been completely modified.

The modifications corresponding to this point have been highlighted in blue.

This manuscript is a resubmission of an earlier submission. The following is a list of the peer review reports and author responses from that submission.

Round 1

Reviewer 1 Report

The paper by Zhdanov and coworkers compared the ERG responses of adults, children, and rabbits using DWT and CWT, and found that the adult signal had higher power than the child signal, and the rabbit signal showed differences in a-wave and b-wave according to the type of response tested, while the Haar Wavelet Found to be advantageous in visualizing frequency components in electrophysiological signals. The data is well organized and solid, the discussion is clear and the conclusion is sound. I recommend publication of this work in Bioengineering.

Author Response

Thank you for taking the time to review our work, and for providing such positive feedback. We greatly appreciate your support and recognition of our efforts. Your comments are truly motivating and we are grateful for your recommendation. Thank you again for your thoughtful review.

Reviewer 2 Report

Authors compared  Human and Rabbit Electroretinogram Based on Time-Domain and Time-Frequency Domain Analysis.

I have the following major concerns

The state of art not fully explored.

Also justification of this comparison could not be seen. 

The results are not compared on standard datasets. Additionally results are not explained well. Difficult to understand what authors wants to convey to their readers.

Discussion section and future work is too weak.

Acceptable

Author Response

The authors would like to thank the reviewer for their valuable time and helpful contributions.

The responses to the Reviewer comments, as well as the corrections made shown below.

1. The ‘state of the art’ is perhaps a strong phrase to use. The study applies DWT and CWT analysis to the ERG waveform – which we acknowledge Gauvin et al’s main contribution to this field. We are aware that the application of signal analysis to the ERG is relatively novel as most clinical applications of the ERG can be evaluated using time-domain parameters of amplitude and peak time. We have added two references that advocate for the expansion of the application of the ERG beyond inherited retinal diseases: Here we have applied a complimentary analysis of the ERG to support its interpretation – in this case with recovery from endophthalmitis in an animal model. Consequently, we have added the following sentence to the introduction to highlight ‘the state of the art’:

The application of signal analysis may provide additional and complimentary information about the underlying neural generators of the ERG and supports the advancement of the use of the ERG as a clinical test of retinal and central nervous system function [55, 56].

[55] Mahroo, Omar A. "Visual electrophysiology and “the potential of the potentials”." Eye (2023): 1-10.

[56] Hamilton, Ruth. "Clinical electrophysiology of vision—commentary on current status and future prospects." Eye 35.9 (2021): 2341-2343.

2. The ISCEV Standard for Full-Field Electroretinography outlines guidelines for conducting and interpreting ERG tests, covering technical specifications, recording parameters, and analysis methods. The updated 2022 version includes new recommendations for electrode types and placements, data analysis software, and normative data for various age groups. It should be noted that changes to the study parameters in Table 1 are permitted as long as the spectral composition remains consistent [1], and the Tomey EP-1000 equipment used for recording electrophysiological signals performs best when following the study protocols detailed in the article. The signals used in the study are published on IEEE DataPort [2].

[1] Robson, Anthony G., et al. "ISCEV Standard for full-field clinical electroretinography (2022 update)." Documenta Ophthalmologica 144.3 (2022): 165-177.

[2] Zhdanov, Aleksei E., et al. ". OculusGraphy: Pediatric and Adults Electroretinograms Database." IEEE DataPort 2020. https://doi.org/10.21227/y0fh-5v04

3. To be more consistent with the study’s main outcome, title of article changed to "Signal analysis of the electroretinogram in a rabbit model of endophthalmitis using discrete and continuous wavelet transforms". The abstract of the article has also been updated.

The subsection "Human and rabbit ERG signals" is supplemented with explanations below.

The differences in the spectral composition of the ERGs between rabbits and humans are of note and may be the result of differences in retinal morphology and may have implications for studies involving rabbit models of retinal pathology [57, 58].

However, in the analysis of electrophysiological signals such as Adult, Child, and Rabbit signals, it has been observed that DWT tends to exhibit low-frequency components due to the presence of software filtering in the electrophysiology equipment used. In contrast, the Haar Wavelet has been found to provide superior visualization of frequency components in such signals.

However, caution should be taken as the features observed using signal analysis show differences to the human ERGs despite a similar shape under photopic and scotopic conditions.

[57] Kondo, M. I. N. E. O. "Animal models of human retinal and optic nerve diseases analysed using electroretinography." Nippon Ganka Gakkai Zasshi 114.3 (2010): 248-78.

[58] Ciulla, Thomas A., et al. "Endothelin-1-mediated retinal artery vasospasm and the rabbit electroretinogram." Journal of ocular pharmacology and therapeutics 16.4 (2000): 393-398.

The following information has been added to the Materials and Methods section.

Article [59] provides an exploration of the evaluation of ophthalmotoxicity stemming from quantum dots based bioconjugates, pertaining to their potential in the treatment of resistant endophthalmitis. This research was conducted using rabbits, with the inclusion of normal rabbit ERG controls.

[59] Ponomarev, Vyacheslav O., et al. "Evaluation of the Ophthalmotoxic Effect of Quantum Dots InP/ZnSe/ZnS 660 and Bioconjugates Based on Them in Terms of the Prospects for the Treatment of Resistant Endophthalmitis. Experimental Research. Part 2 (Stage 1) /ZnS 660 and bioconjugates based on them in terms of prospects for the treatment of resistant endophthalmitis. Experimental study. Part 2 (1st stage)." Ophthalmology in Russia (2021): 876-884.

Reviewer 3 Report

In this paper, the authors try to analyze and compare features of ERG waveform from different subjects. There were sevieal deficiencies in the study.

1. Study aim is not clear. The title is “Comparison Between Human and Rabbit Electroretinogram”. While in abstract, the authors discribed “The primary aim of this study was to monitor and evaluate the effects of treatment in a New Zealand rabbit model of endophthalmitis via electroretinogram waveform analysis.” But the results compared adult, child, and rabbit electroretinogram responses. The authors should claerify the study aim and design the content more reasonably.

2. There were similar study published (PMID: 25061605, PMID: 27562839). The authors should think more about their novelty.

3. The study involved human and aninal subjects. Did any animal care and use committee approved the rabbit study?

4. Only two adults and two children ERG result were collected. There were no any demographics information of these human subjects. The small sample size with no any statistical analyses could not get a convincing result.

5. For endophthalmitis rabbit model, the study should include normal rabbit control, endophthalmitis rabbit model, and endophthalmitis plus treatment groups. Therfore one can evaluate the effects of treatment in rabbit model of endophthalmitis.

Author Response

The authors would like to thank the reviewer for their valuable time and helpful contributions.

The responses to the Reviewer comments, as well as the corrections made shown below.

1. To be more consistent with the study’s main outcome, title of article changed to "Signal analysis of the electroretinogram in a rabbit model of endophthalmitis using discrete and continuous wavelet transforms". The abstract of the article has also been updated.

The subsection "Human and rabbit ERG signals" is supplemented with explanations below.

The differences in the spectral composition of the ERGs between rabbits and humans are of note and may be the result of differences in retinal morphology and may have implications for studies involving rabbit models of retinal pathology [57, 58].

However, in the analysis of electrophysiological signals such as Adult, Child, and Rabbit signals, it has been observed that DWT tends to exhibit low-frequency components due to the presence of software filtering in the electrophysiology equipment used. In contrast, the Haar Wavelet has been found to provide superior visualization of frequency components in such signals.

However, caution should be taken as the features observed using signal analysis show differences to the human ERGs despite a similar shape under photopic and scotopic conditions.

[57] Kondo, M. I. N. E. O. "Animal models of human retinal and optic nerve diseases analysed using electroretinography." Nippon Ganka Gakkai Zasshi 114.3 (2010): 248-78.

[58] Ciulla, Thomas A., et al. "Endothelin-1-mediated retinal artery vasospasm and the rabbit electroretinogram." Journal of ocular pharmacology and therapeutics 16.4 (2000): 393-398.

2. The article (PMID: 25061605) compares various methods for analyzing human photopic ERGs, such as time domain analysis, Fourier analysis, and wavelet transforms. The study was conducted on 40 healthy subjects (26 females and 14 males, 21.5 - 38.3 years old) and found that the discrete wavelet transform method was the most effective, allowing for the extraction of more relevant descriptors of the ERG signal, and facilitating follow-up of disease progression.

The study (PMID: 27562839) compared measurements of the photopic negative response (PhNR) obtained through time and time-frequency domain analyses in 20 healthy subjects (aged 24-65 years). The authors found that all three metrics (amplitude at the PhNR trough, amplitude at 72 ms following stimulus onset, and energy in the 11 Hz, 60-120 ms DWT frequency bin) provided similar estimates of the PhNR.

It should be noted that the article does not provide a universal method, nor does it use the signals of unhealthy subjects or laboratory animals. Moreover, in both articles, different electrophysiological research protocols were used, separate from our article. Thus, the scientific novelty of this study lies in the comparison of healthy and unhealthy subjects (children, adults, rabbits) in the time-frequency domain using the study protocol Maximum 2.0 ERG Responses and Scotopic 2.0 ERG Responses.

3. The following information has been added to the Materials and Methods section.

The research involving laboratory animals was conducted in strict adherence to the principles outlined in the code of ethics, specifically the "Directive 2010/63/EU of the European Parliament and the Council of the European Union".

The Institutional Review Board Statement states as follows.

The study was approved by the ethics committee of Ural Federal University Named after the First President of Russia B. N. Yeltsin (Conclusion No. 1 dated 1 February 2021).

4. The database used in the study includes five types of adult and pediatric ERGs: Scotopic 2.0 ERG Response (53 pediatric signals, 23 adult signals), Maximum 2.0 ERG Response (80 pediatric signals, 42 adult signals), Photopic 2.0 ERG Response (74 pediatric signals, 32 adult signals), and Photopic 2.0 EGR Flicker Response (63 pediatric signals, 38 adult signals). 20 signals of Scotopic 2.0 ERG Oscillatory Potentials in addition to that. The entire statistical analysis of the database is given in the article [1]. This article is given in the References section at number 54 and the Data Availability Statement [2]. Since the statistical analysis of the database used is not the research subject, the team of authors considers it sufficient to mention the relevant publications in the article.

It should be noted that for the study, the signals most qualitatively demonstrating healthy and unhealthy subjects were selected.

[1] Zhdanov, Aleksei, et al. "Advanced Analysis of Electroretinograms Based on Wavelet Scalogram Processing." Applied Sciences 12.23 (2022): 12365.

[2] Zhdanov, Aleksei E., et al. ". OculusGraphy: Pediatric and Adults Electroretinograms Database." IEEE DataPort 2020. https://doi.org/10.21227/y0fh-5v04

5. Thank you for your comment and feedback. We agree that including normal rabbit control, endophthalmitis rabbit model, and endophthalmitis plus treatment groups is important in evaluating the effects of treatment in a rabbit model of endophthalmitis. However, in this study, we chose to focus on ERG's comparison based on wavelet analysis. We believe that Figure 7 demonstrates the possibility of using wavelet analysis for diagnosis and evaluation of therapeutic effects. While we understand the importance of including normal rabbit controls, it was not our primary focus in this study. Furthermore, we are unable to publish all the signals of rabbits as they are part of a study dedicated to the development of new ophthalmic drugs based on quantum dots [1, 2].

[1] Ponomarev, Vyacheslav O., et al. "Evaluation of the ophthalmotoxic effect of quantum dots and bioconjugates based on them in terms of the prospects for the treatment of resistant endophthalmitis. Experimental Research (Stage 1)." Ophthalmology in Russia 18(3) (2021):476-487.

[2] Ponomarev, Vyacheslav O., et al. "Evaluation of the Ophthalmotoxic Effect of Quantum Dots InP/ZnSe/ZnS 660 and Bioconjugates Based on Them in Terms of the Prospects for the Treatment of Resistant Endophthalmitis. Experimental Research. Part 2 (Stage 1) /ZnS 660 and bioconjugates based on them in terms of prospects for the treatment of resistant endophthalmitis. Experimental study. Part 2 (1st stage)." Ophthalmology in Russia (2021): 876-884.

The following information has been added to the Materials and Methods section.

Article [59] provides an exploration of the evaluation of ophthalmotoxicity stemming from quantum dots based bioconjugates, pertaining to their potential in the treatment of resistant endophthalmitis. This research was conducted using rabbits, with the inclusion of normal rabbit ERG controls.

[59] Ponomarev, Vyacheslav O., et al. "Evaluation of the Ophthalmotoxic Effect of Quantum Dots InP/ZnSe/ZnS 660 and Bioconjugates Based on Them in Terms of the Prospects for the Treatment of Resistant Endophthalmitis. Experimental Research. Part 2 (Stage 1) /ZnS 660 and bioconjugates based on them in terms of prospects for the treatment of resistant endophthalmitis. Experimental study. Part 2 (1st stage)." Ophthalmology in Russia (2021): 876-884.
